# AOT: Appearance Optimal Transport Based Identity Swapping for Forgery Detection

**Hao Zhu** [1,2,*] **Chaoyou Fu** [1,3,*], **Qianyi Wu** [4], **Wayne Wu** [4], **Chen Qian** [4], **Ran He** [1,3†]

[1] NLPR & CEBSIT & CRIPAC, CASIA [2] Anhui University
[3] University of Chinese Academy of Sciences [4] SenseTime Research
haozhu96@gmail.com {chaoyou.fu,rhe}@nlpr.ia.ac.cn
{wuqianyi,wuwenyan,qianchen}@sensetime.com

## Abstract

Recent studies have shown that the performance of forgery detection can be improved with diverse and challenging Deepfakes datasets. However, due to the lack of Deepfakes datasets with large variance in appearance, which can be hardly produced by recent identity swapping methods, the detection algorithm may fail in this situation. In this work, we provide a new identity swapping algorithm with large differences in appearance for face forgery detection. The appearance gaps mainly arise from the large discrepancies in illuminations and skin colors that widely exist in real-world scenarios. However, due to the difficulties of modeling the complex appearance mapping, it is challenging to transfer fine-grained appearances adaptively while preserving identity traits. This paper formulates appearance mapping as an optimal transport problem and proposes an Appearance Optimal Transport model (AOT) to formulate it in both latent and pixel space. Specifically, a relighting generator is designed to simulate the optimal transport plan. It is solved via minimizing the Wasserstein distance of the learned features in the latent space, enabling better performance and less computation than conventional optimization. To further refine the solution of the optimal transport plan, we develop a segmentation game to minimize the Wasserstein distance in the pixel space. A discriminator is introduced to distinguish the fake parts from a mix of real and fake image patches. Extensive experiments reveal that the superiority of our method when compared with state-of-the-art methods and the ability of our generated data to improve the performance of face forgery detection.

## 1 Introduction

Face forgery detection refers to detect whether a given image has been altered. However, recent detection baselines are data-driven [42, 39, 34, 11, 1], and the lack of Deepfake datasets with large differences in appearance may render these detection algorithms ineffective in this situation. In this work, we provide a new identity swapping algorithm with large differences in appearance to improve the robustness of face forgery detection methods.

Recently, many identity swapping works have been devoted to addressing the recombination difficulty of identities and attributes in the source and the target faces. Earlier works either directly warp a source face to a target pose [16] or leverage 3D models to flexibly align the source face with a target video [10, 36, 43]. More recently, benefiting from the success of deep learning, further identity swapping progress has been achieved [30, 41, 44, 32].

---

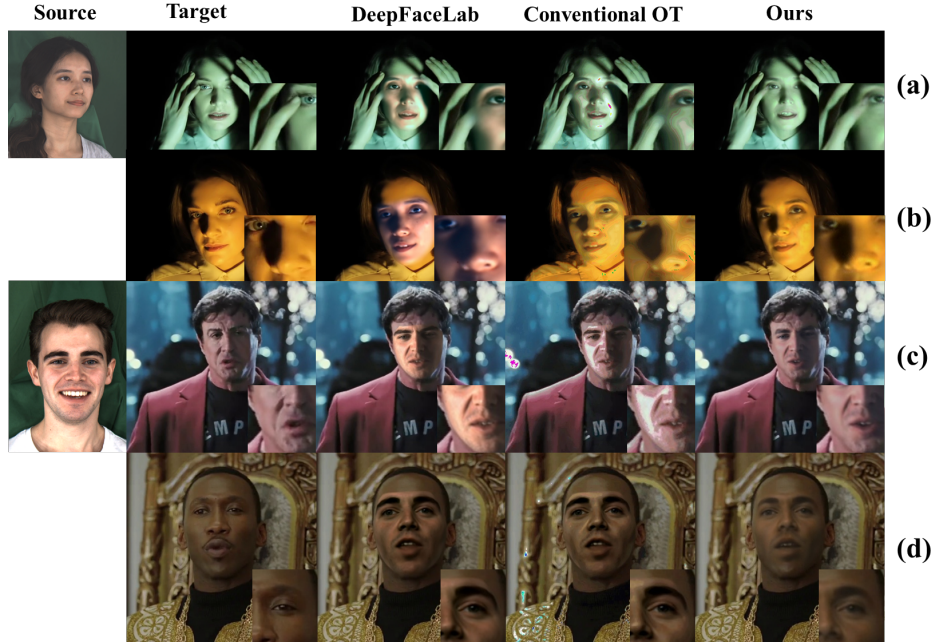

| Source | Target | DeepFaceLab | Conventional OT | Ours | |
|---|---|---|---|---|---|
| | | | | | (a) |
| | | | | | (b) |
| | | | | | (c) |
| | | | | | (d) |

Figure 1: Comparisons with DeepFaceLab [46] and conventional optimal transport [15]. These two methods degrade significantly under complex appearance conditions. Obvious artifacts and inconsistent appearances with the target faces are observed (the third and the fourth columns). In contrast, our proposed AOT achieves more realistic results in various situations.

However, the appearance gaps between the source and target faces, as a challenging and essential problem in identity swapping, have rarely been studied. As presented in Fig. 1 (the first and the second columns), the appearance gaps mainly arise from large discrepancies in skin colors and illuminations. Unfortunately, existing identity swapping methods often degrade significantly when facing such an intractable case, as exemplified in Fig. 1 (the third column).

Natural appearances transferred results can help researchers investigate a powerful forgery detection method. However due to the difficulties of modeling the precise mapping for extremely complex appearances, it is challenging to transfer fine-grained appearances with preservation of identity traits, and the swapped images tend to have obvious artifacts and inaccurate skin colors or illuminations. To achieve a vivid appearance transferred result, we consider finding a map between color histogram pairs with minimum cost. By manipulating the histogram distribution, i.e., mapping the histogram distribution of the source image to that of the target, we can not only capture color differences but also preserve structures of the source image. Therefore, optimal transport (OT) is a natural idea that aims to solve a transport plan $\gamma$ to map one probability distribution to another with minimal cost [27].

Nevertheless, the conventional optimal transport plan is solved via optimization-based methods [15, 19], which makes it challenging to directly apply the OT to identity swapping due to the following facts. (1) Calculating OT between images with unbalanced pixel histograms may derive a mismatching mapping and further result in a discontinuous synthesized image (see the fourth column of Fig. 1 (a, b)). (2) Facial geometry information is neglected during the transport process, which makes it difficult to perform a fine-grained appearance translation (see the fourth column of Fig. 1 (c, d)). (3) The heavy computational burden of OT restricts its application in large-scale identity swapping.

In this paper, we propose a novel way, named as Appearance Optimal Transport model (AOT), which provides a new method to generate data for face forgery detection. AOT aims to tackle the appearance gaps for identity swapping, We formulate the appearance mapping as an OT problem and formulate it in both latent space and pixel space. We first propose a two-branch perceptual encoder as shown in Fig. 2. The feature encoder encodes the images into the high-level features. In addition, corresponding coordinates and normals are extracted from the 3D fitting model to present facial geometry and lighting orientational information [56], respectively. Furthermore, a Neural Optimal Transport Plan Estimation (NOTPE) is designed to solve the OT problem by minimizing

the Wasserstein distance (WD) of the features in the latent space, and it enables more continuous synthesis and reduced computation. Subsequently, the relighting generator decodes the features into the pixel space and further refines the results via a segmentation game [68, 54]. A discriminator is introduced to predict the real parts from the real-fake mixture image blocks, while the relighting generator attempts to disturb the discriminator via generating more realistic images. By this approach, we force the pixel space of the synthesized image and that of the target image to be as similar as possible. Finally, it is worth emphasizing that the proposed AOT can be flexibly applied to existing identity swapping methods [46, 44] to improve their performance. This means that our model can produce diverse and realistic Deepfakes to improve the performance of detection algorithms. Our contributions can be summarized as follows:

- We formulate the rather challenging appearance transfer in identity swapping as an optimal transport problem and formulate it via the proposed AOT. It can be flexibly applied to recent identity swapping methods and thus produce various results. To the best of our knowledge, we are the first to thoroughly study the appearance gap problem in identity swapping.

- We propose to tackle optimal transport in the latent space via neural optimal transport plan estimation, enabling better performance and reduced computation.

- We develop a segmentation game to further minimize the Wasserstein distance in the pixel space, forcing the generator to synthesize photorealistic results.

- Extensive experiments on FF++ [53] and DPF-1.0 [25] demonstrate the superiority of our method over state-of-the-art identity swapping methods [46, 44] in the situation of large appearance gaps and the ability of our generated data to improve face forgery detection methods.

## 2  Related Work

### 2.1  Identity Swapping and Forgery Detection

Earlier identity swapping methods are mainly based on 3D models [6, 7, 26]. Cheng et al. [10] propose an approach based on the 3D morphable model (3DMM) [6] and an expression database. Lin et al. [36] eliminate the limitations of pose and appearance similarity with a refined texture. Nirkin et al. [43] use a 3DMM and segmentation to transfer the face area seamlessly. Recently, learning-based works have improved replacement quality. Korshunova et al. [30] leverage image-to-image translation for identity swapping. Natsume et al. [41] extract the features of the face region and the non-face region via a deep neural network and combine these features of different identities to synthesize the swapped images. Nirkin et al. [44] propose a subject-agnostic pipeline that contains face reenactment, face inpainting, and identity swapping. Li et al. [32] implement occlusion aware identity swapping via adaptive attentional denormalization. However, the appearance gaps between faces have rarely been addressed. Recently, many face forgery detection benchmarks have been proposed. The Face Forensics Benchmark [53] includes six image level face forgery detection benchmarks [50, 20, 14, 11, 1, 5]. Celeb-DF [35] also provides seven methods [70, 66, 42, 39, 34, 11, 1] for training and testing on different datasets.

### 2.2  Image Harmonization and Photorealisic Style Transfer

Transferring the appearance from one image to another one requires learning good matching between them. Earlier works attempt to solve this problem by transferring statistics of images [48, 51] or exploiting gradient-domain information [45, 57]. However, these works directly learn the matching in pixel space, ignoring the authenticity of the output image. Recently, Zhu et al. [72] leverage convolutional neural networks (CNNs) to assess the realism of the photo. Tsai et al. [61] are the first to employ end-to-end deep convolutional networks to learn contextual and semantic information. Cong et al. [12] propose a domain verification discriminator to verify the domain of the foreground and the background. Instead of the appearance, some works learn to transfer a higher-level style between images with structural preservation [38, 33, 67, 3]. Luan et al. [38] introduce a smoothness-based loss term to constrain the transformation from the input to the output. Li et al. [33] propose a smoothing step followed by the stylization step to enforce the spatial inconsistency. Yoo et al. [67] propose a wavelet corrected transfer based on whitening and coloring transforms for style transfer with structural information preservation. However, these works can only transfer the global style with

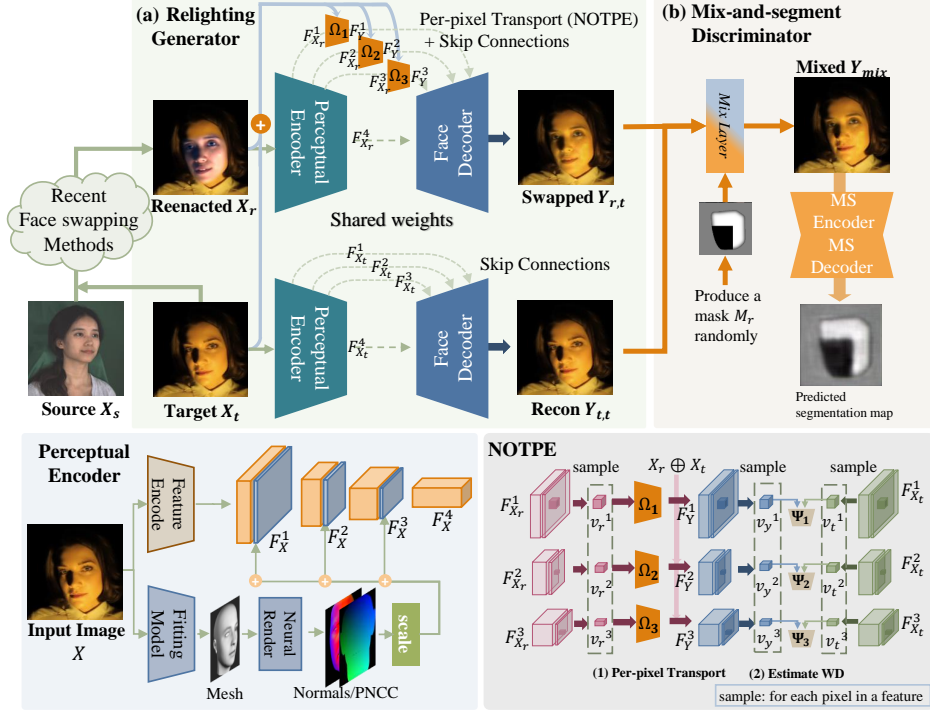

Figure 2: The pipeline of the AOT that involves two main models: a relighting generator and a mix-and-segment discriminator. First, the reenacted face and the target face are fed to the perceptual encoder to obtain the multiscale features for solving the optimal transport plan. Subsequently, we decode these features into images and then mix them via a randomly produced mix-mask. Finally, a mix-and-segment discriminator is introduced to assess the realness of patches.

structure preservation, while the fine-grained appearance translation cannot be achieved. This issue limits their applications in identity swapping.

## 2.3 Optimal Transport

The optimal transport (OT) problem is first formulated by Monge [40] to study the optimal transportation and resource allocation problem. It aims to solve a transport mapping $T(\cdot)$ from space $\mathbb{P}_s$ to space $\mathbb{P}_t$ as:

$$T^\star = \operatorname*{argmin}_{T(X) \sim \mathbb{P}_t} \mathbb{E}_{X \sim \mathbb{P}_s}[c(X, T(X))], \tag{1}$$

where $c(\cdot)$ denotes the cost function. However, Monge's formulation is nonconvex, and there is no guarantee of finding a solution. This can be improved by adopting Kantorovich's formulation, which is a relaxation of Eq. (1):

$$\gamma^\star = \inf_{\gamma \in \prod(\mathbb{P}_s, \mathbb{P}_t)} \mathbb{E}_{(x,y) \sim \gamma}[c(x - y)]. \tag{2}$$

The Kantorovich problem aims to solve the joint distribution $\gamma^\star$ of $\mathbb{P}_s \times \mathbb{P}_t$ rather than the mapping $T^\star$, and the solution is guaranteed to be feasible. Due to the more useful properties, many works based on Kantorovich relaxation have emerged, such as color transfer [21, 49, 56], image super-resolution [29], domain adaption [13, 9], and generative models [4, 23, 37, 2]. However, the OT map has historically been difficult to compute [47].In addition, the continuity and realism of the produced results may be neglected by directly applying OT between images, which restricts its application in identity swapping. Therefore, we formulate appearance mapping as an optimal transport problem and formulate it in both latent and pixel space via neural networks.

## 3 Approach

In this section, we elaborate our AOT as shown in Fig. 2. AOT involves two main models, a relighting generator (RG) and a mixed-and-segment discriminator (MSD), to transfer the appearance in the

latent space and the pixel space, respectively. AOT is trained in a GAN [22] manner. First, a reenacted face $X_r$ produced by recent identity swapping methods [46, 44], and a target face $X_t$, are fed to the perceptual encoder to produce latent features $F_{X_r}^i$ and $F_{X_t}^i$, which indicate $i$-th level latent features of $X_r$ and $X_t$, respectively. A Neural Optimal Transport Olan Estimation (NOTPE) is then applied to each patch of $F_{X_r}^i$ to produce the transported patch of $F_Y^i$. Finally, a face decoder is incorporated to predict $Y_{r,t}$. $Y_{r,t}$ should maintain the identity traits (facial attributes and geometry) of the reenacted face $X_r$ and the appearance (skin colors and illuminations) of the target face $X_t$.

To further exploit the Wasserstein distance in the pixel space, a mix-and-segment discriminator (MSD) is introduced. $Y_{t,t}$, $Y_{r,t}$, and the randomly generated mix-mask $M_r$ are fed to the mix layer to produce the mixed image $Y_{mix}$. Subsequently, the MSD predicts the real part from the mixed patches, forcing the generator to synthesize photorealistic results.

## 3.1 Relighting Generator

### 3.1.1 Theoretical Derivation

As illustrated in Sec. 1 that in a complex high-contrast appearance condition, the color histogram distributions of two faces are usually unbalanced, which may cause ambiguous appearance translation without the guidance of extra position information. Therefore, it is intractable to directly apply the optimal transport plan $\gamma$ to the histogram of image color space (3 dimensions, e.g., RGB color space) for precise appearance translation.

To tackle these issues, we leverage a full convolutional network to encode pixel space into the smooth latent space ($k$ dimensions). As convolution networks preserve the spatial structure of the input, we build up a mixed histogram ($k + 6$ dimensions) by introducing the per-pixel PNCC feature (3 dimensions) [73] and normals (3 dimensions). The solved map is preferred to mapping the histogram with similar geometry and orientation information.

Formally, we introduce a vector $v = (f, x, n)$ that concatenates the feature, position, and normal of a pixel. Our core problem here is to find an optimal transport plan between the histogram of reenacted $v_r$ and target $v_t$ features under the condition of $X_r$ and $X_t$. We rewrite Eq. (2) as:

$$W = \underset{\gamma \in \prod(\mathbb{V}_r, \mathbb{V}_t | X_r, X_t)}{\operatorname{argmin}} \mathbb{E}_{(v_r, v_t) \sim \gamma}[c(v_r, v_t)]. \tag{3}$$

This can be rewritten from the Kantorovich-Rubinstein duality [28] by introducing a 1-Lipschitz function $\Psi$:

$$W = \sup_{\Psi \in \mathcal{F}^1} [\mathbb{E}_{v_r \sim \mathbb{V}_r}(\Psi(v_r)) - \mathbb{E}_{v_t \sim \mathbb{V}_t}(\Psi(v_t))]. \tag{4}$$

However, it is highly intractable to solve $\gamma$ between all continuous distributions. A push-forward method is therefore involved by leveraging a conditional mapping function $\Omega(\cdot)$ that transports $v_r^i \in F_{X_r}^i$ to $v_t^i \in F_{X_t}^i$ implicitly [31], which can be formed as a minimax problem:

$$\min_{\Omega} \max_{\Psi \in \mathcal{F}^1} \mathbb{E}[\Psi(v_t)] - \mathbb{E}[\Psi(\Omega(v_r | X_r, X_t))]. \tag{5}$$

Inspired by the fact that neural networks can be used to fit any functions theoretically, $\Omega(\cdot)$ and $\Psi(\cdot)$ are represented by two neural networks parameterized by $\theta$ and $\omega$, respectively. In addition, weight clipping is adopted to enforce the 1-Lipschitz constraint.

### 3.1.2 Network Design

Specifically, we design a perceptual encoder that produces four different features. The first three features concatenated with scaled PNCC and normals represent the attributes, while the last feature represents the identity information. The PNCC and normals are extracted via 3DDFA [73]. We feed $X_r$ and $X_t$ to the perceptual encoder and yield $F_{X_r}^{1\sim4}$ and $F_{X_t}^{1\sim4}$, respectively.

To transport the attribute feature pixels $v_r^i \in F_{X_r}^i$ to the target distribution, we introduce a Neural Optimal Transport Plan Estimation (NOTPE) strategy as shown in Fig. 2. Each feature $v_r^i \in F_{X_r}^i$ along with $X_r$ and $X_t$ is inputted to $\Omega^i$ (3 fully connected layers for each) to obtain the transported pixel $v_y^i \in F_Y^i$:

$$v_y^i = \Omega^i(v_r^i | X_r, X_t). \tag{6}$$

Meanwhile, three $\Psi^{i=1\sim3}$ (2 fully connected layers for each) are proposed to estimate the Wasserstein distance of the feature distribution.

We apply an alternating stochastic gradient algorithm to solve Eq. (5): in each iteration, we first perform $d$ steps of gradient ascent on $\Omega^i$ with $v_t^i$ (fixing $\Psi^i$), respectively, and then fix $\Omega^i$ to update $\Psi^i$. Rewriting Eq. (5), the training loss of NOTPE can be summarized as:

$$\mathcal{L}_{NOTPE} = \sum_{i=1}^{3} \min_{\Omega^i} \max_{\Psi^i \in \mathcal{F}^1} \mathbb{E}[\Psi^i(\Omega^i(v_r^i|X_r, X_t))] - \mathbb{E}[\Psi^i(v_t^i)]. \tag{7}$$

After obtaining the transported features $F_Y^i$, a face decoder is introduced to decode the attribute features along with the identity feature into the pixel space. The decoder contains three transpose convolutional layers. The identity feature is fed into an upsampling network, and the attribute features are fed to the different upsampling networks via skip-connections.

To capture the content and style from $X_r$ and $X_t$, respectively, $\mathcal{L}_{cont}$ and $\mathcal{L}_{appear}$ are involved. Furthermore, an additional reconstruction loss $\mathcal{L}_{recon}$ is applied between $X_t$ and $Y_{t,t}$:

$$\mathcal{L}_{RG} = \underbrace{\| vgg(Y_{r,t}) - vgg(X_r) \|_1}_{\mathcal{L}_{cont}} + \underbrace{\| Y_{r,t} - X_t \|_1}_{\mathcal{L}_{appear}} + \underbrace{\| Y_{t,t} - X_t \|_1}_{\mathcal{L}_{recon}}. \tag{8}$$

### 3.2 Mix-and-Segment Discriminator

As the correlation between high-level and low-level of the same image, we also bridge the appearance gaps in the low-level pixel space. Due to the characteristics of identity swapping, $X_r$, $X_t$, and $Y_{r,t}$ should have the same pose and expression. Furthermore, $X_t$ and $Y_{r,t}$ are required to have the same color style. Although there are subtle structural differences between $Y_{r,t}$ and $X_t$, the colors in the same position of $Y_{r,t}$ and $X_t$ are supposed to be similar in most areas. Therefore, if we randomly mix a small patch of $Y_{r,t}$ and $X_t$, it should be difficult to find the mixed patches when $Y_{r,t}$ is well blended [68, 54].

Extended from WGAN [4], which minimizes the Wasserstein distance between two images, we introduce MSD to predict the pixelwise Wasserstein distance in $Y_{mix}$. To achieve this, we randomly produce a mix-mask of $M_r$ ( $0 \sim 1$), and the mixed face $Y_{mix}$ is calculated as:

$$Y_{mix} = (1 - M_r) \times Y_{r,t} + M_r \times X_t. \tag{9}$$

By feeding $Y_{mix}$ to the MSD, a variant of U-Net [52], we obtain a map of the pixelwise Wasserstein distance. The MSD is updated with a minimax problem:

$$\mathcal{L}_{MSD} = \min_{G} \max_{MSD \in \mathcal{F}^1} \mathbb{E}[\underbrace{(1 - M_r) \times MSD(Y_{mix})}_{fake\ part}] - \mathbb{E}[\underbrace{M_r \times MSD(Y_{mix})}_{real\ part}]. \tag{10}$$

Finally, the AOT is trained with a weighted sum of the above losses:

$$\mathcal{L}_{total} = \lambda_1 \mathcal{L}_{cont} + \lambda_2 \mathcal{L}_{appear} + \lambda_3 \mathcal{L}_{recon} + \lambda_4 \mathcal{L}_{MSD}, \tag{11}$$

where the $\lambda_i$ denotes the weight of each loss.

## 4 Experiments

We evaluate our model on prevalent benchmark datasets: FaceForensics++ (FF++) [53] and DeeperForensics-1.0 (DPF-1.0) [25]. The former is an in-the-wild dataset, and the latter is collected under challenging lighting conditions. Our experiments are conducted based on two representative identity swapping methods: DeepfaceLab (DFL) [46], which requires to retrain the model for different source identities, and FSGAN [44], a landmark-guided subject agnostic method.

We also compare our method with related image harmonization or appearance transfer methods such as Poisson blending [45], deep image harmonization (DIH) [60], optimized-based style transfer (STHP) [55], and learning-based photorealistic style transfer (WCT$^2$) [67]. These works aim at transferring the color or style with structure preservation. This goal is in line with ours. Please refer to the Appendix for more details about network designation, training strategies, and data preparation.

## 4.1 Qualitative Results

As shown in Fig. 3, we first compare our model with the original identity swapping [46, 44] integrated blending algorithm on DPF-1.0 [25]. It proves that our AOT is able to narrow the appearance gaps under complex lighting directions. Next, we further compare the AOT with related appearance transfer or blending methods. As shown in Fig. 4, these methods not only fail to transfer the appearances of the targets but also cannot effectively eliminate the obvious boundaries. Specifically, Poisson blending [45] and DIH [60] tend to cause ghosting, while STHP [55] and WCT$^2$ [67] often lead to inappropriate global or local appearances.

## 4.2 Quantitative Results

As reported in Table 1, quantitative experiments are conducted on both FF++ and DPF-1.0 with DFL [46] and FSGAN [44], as well as related methods [45, 60, 55, 67].

### 4.2.1 Metrics Evaluation

Motivated by [3], to evaluate generation quality and style accuracy between the generated results and target faces, we calculate the SSIM [64] (the higher the better) and Gram matrix loss (the lower the better). Furthermore, to evaluate the geometric consistency, we first extract the edges via HED [65] and then calculate the SSIM between the edges of the generated result and those of the source image. For the sake of clarity, we name the SSIM on images as SSIM-whole and the SSIM on edges as SSIM-edge. As reported in Table 1 (a), the AOT achieves the best performance on both the SSIM-image and Gram matrix, and achieves comparable results on SSIM-edge. This result verifies that the AOT is capable of accurately transferring the appearance with geometric consistency.

### 4.2.2 User Evaluation

A user study is also conducted on FF++ [53] and DPF-1.0 [25] with respect to both realism and appearance. We recruit 31 volunteers: given video pairs (ours and results of other methods), the volunteers are asked to answer the following questions: 1) Which one looks more like an unaltered video? and 2) Which one better preserves the appearance of the target video? As reported in Table 1 (b), we achieve the highest score for both realism and appearance (the scores in the table represent the rate at which users picked the result of the method), especially on DPF-1.0, which contains the challenging lighting conditions.

## 4.3 Analysis of Components

**Relighting Generator.** Firstly, we remove the 3D features (PNCC and normals): the performance drops (Table 2 (a)), and the results tend to transfer the appearance with incorrect positions (Fig. 5 (a)). In addition, NOTPE is essentially a feature transfer or a feature fusion component. To assess the its effectiveness, we compare it with other related components such as ADD (add pixel-wise), CONCAT (concatenate channel-wise), and AdaIN [24]. As reported in Table 2 (Feature Transfer), both 'ADD' and 'CONCAT' have a low score on SSIM-edge but also result in a high score on Gram loss. This means that simply fusing two features cannot address this problem well, as shown in Fig. 5 (b,c,d). By leveraging AdaIN [24], we get better performance. However, the results of AdaIN only preserve

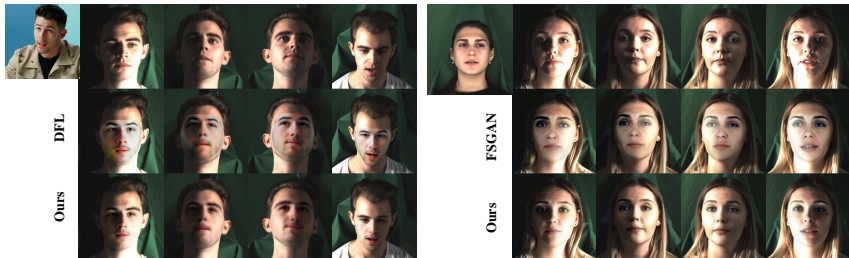

Figure 3: Comparisons with DFL [46] (left) and FSGAN [44] (right) on DPF-1.0 [25] under different lighting directions.

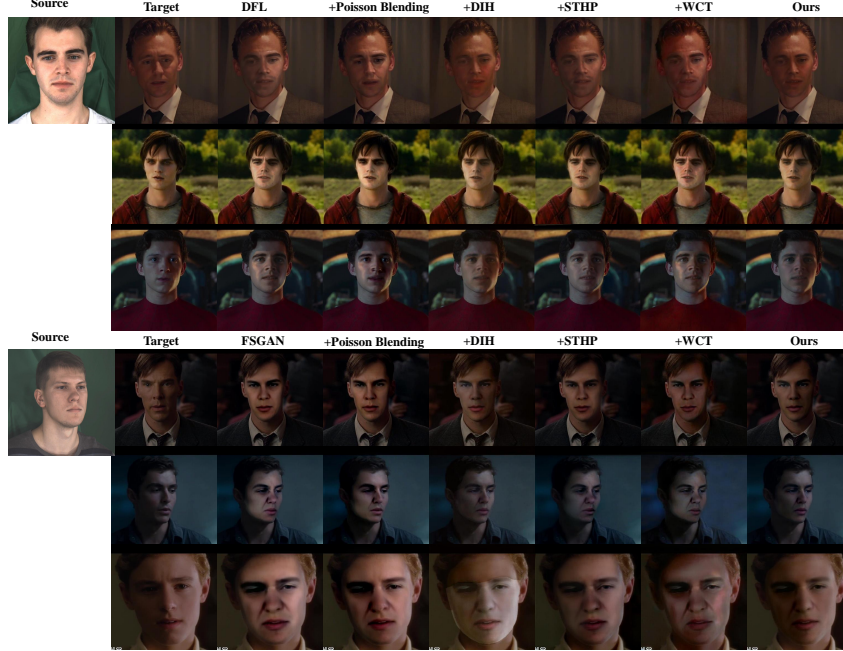

Figure 4: Comparisons with related blending, harmonization, and color transfer methods based on DFL [46] (top) and FSGAN [44] (bottom). Our method better preserves the appearance of the target.

Table 1: Quantitative results and user study (↑: the higher the better; ↓: the lower the better).

| | (a) Metric Evaluation | | | | | | (b) User Evaluation | | | |
| | SSIM-edge ↑ | | SSIM-whole ↑ | | Gram loss ↓ | | Realism ↑ | | Appearance ↑ | |
| | FF++ | DPF-1.0 | FF++ | DPF-1.0 | FF++ | DPF-1.0 | FF++ | DPF-1.0 | FF++ | DPF-1.0 |
|---|---|---|---|---|---|---|---|---|---|---|
| DFL | 0.8496 | 0.8239 | 0.7331 | 0.7153 | 0.005324 | 0.013540 | 0.189679 | 0.149041 | 0.086139 | 0.131675 |
| DFL+PB | 0.7922 | 0.7711 | 0.7192 | 0.6979 | 0.006172 | 0.008128 | 0.008368 | 0.006575 | 0.029757 | 0.027798 |
| DFL+DIH | 0.8053 | 0.8084 | 0.7463 | 0.7182 | 0.003123 | 0.006225 | 0.078103 | 0.061370 | 0.118246 | 0.110461 |
| DFL+WCT$^2$ | 0.8252 | 0.7825 | 0.7022 | 0.7193 | 0.002849 | 0.003598 | 0.207113 | 0.162740 | 0.138606 | 0.129481 |
| DFL+STHP | 0.7903 | 0.8163 | 0.7082 | 0.7385 | 0.005022 | 0.009185 | 0.091353 | 0.071781 | 0.094753 | 0.088515 |
| DFL+FR (Ours) | **0.8828** | **0.8301** | **0.8022** | **0.7810** | **0.002011** | **0.003578** | **0.425384** | **0.548493** | **0.532498** | **0.512070** |
| FSGAN | 0.8298 | **0.7487** | 0.7395 | 0.6887 | 0.007765 | 0.010041 | 0.210364 | 0.191358 | 0.039322 | 0.156914 |
| FSGAN+PB | 0.8076 | 0.7298 | 0.7324 | 0.6621 | 0.007872 | 0.097240 | 0.019124 | 0.017396 | 0.035081 | 0.033917 |
| FSGAN+DIH | 0.8119 | 0.7224 | 0.7431 | 0.6511 | 0.005546 | 0.006777 | 0.093152 | 0.084736 | 0.158057 | 0.071562 |
| FSGAN+WCT$^2$ | 0.8219 | 0.7304 | 0.7109 | 0.6742 | 0.002712 | 0.003972 | 0.109192 | 0.099327 | 0.146492 | 0.069698 |
| FSGAN+STHP | 0.8397 | 0.7333 | 0.7392 | 0.6408 | 0.008114 | 0.008793 | 0.062307 | 0.056678 | 0.042791 | 0.041372 |
| FSGAN+FR (Ours) | **0.8401** | 0.7446 | **0.7780** | **0.7309** | **0.002117** | **0.003674** | **0.505861** | **0.550505** | **0.578258** | **0.626537** |

attributes of the target but lost identity information of the source. It is clear that the AOT achieves the best performance. Furthermore, Fig. 6 visualizes the feature transportation of different layers by applying PCA to reduce the dimensions to 2-dimensional vectors, which proves that NOTPE plays a crucial role in OT transport plan estimation.

**Mix-and-Segment Discriminator.** To verify the optimal transport in pixel space, both qualitative and quantitative results are presented in Fig. 5 and Table 2 (Discriminator), respectively. After removing the MSD or replacing it with the original discriminator proposed in [22], the SSIM-whole

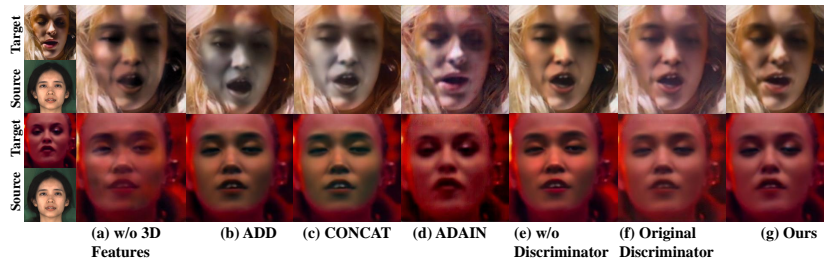

(a) w/o 3D Features    (b) ADD    (c) CONCAT    (d) ADAIN    (e) w/o Discriminator    (f) Original Discriminator    (g) Ours

Figure 5: Qualitative results of the ablation study.

Table 2: Component analysis on DPF-1.0.

| | Settings | SSIM-edge ↑ | SSIM-whole ↑ | Gram loss ↓ |
|---|---|---|---|---|
| Feature Transfer | (a) w/o 3D features | 0.7736 | 0.7183 | 0.019993 |
| | (b) Add | 0.7903 | 0.7196 | 0.036263 |
| | (c) Concat | 0.7883 | 0.7265 | 0.036814 |
| | (d) AdaIN | 0.8026 | 0.7522 | 0.007700 |
| Discriminator | (e) w/o D | 0.8065 | 0.7255 | 0.016075 |
| | (f) Original D | 0.7998 | 0.7154 | 0.019354 |
| | (g) Full | **0.8301** | **0.7810** | **0.003578** |

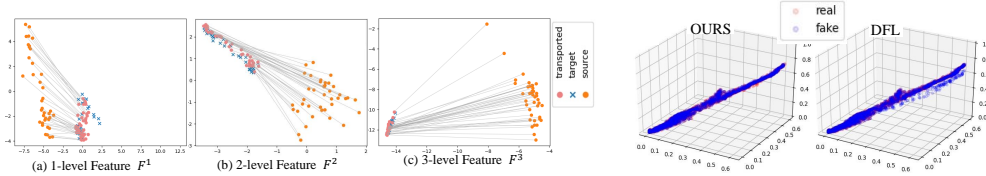

Figure 6: The visualization of the feature mapping of $F^1$, $F^2$, and $F^3$ (zoom in for more details).

Figure 7: Visualization of image pixel distribution on DFL [46].

and Gram loss decrease. The subtle color differences also can be noticed, as shown in Fig. 5 (e,f). We further visualize the differences of pixel distributions with or without the MSD, as shown in Fig. 7. By adding the MSD, the distribution of generated pixels is closer to that of real pixels, which verifies that the proposed MSD is capable of capturing the slight color differences.

## 4.4 Failure Case

Occlusion is a challenging problem in identity swapping. Therefore, if the prestage identity swapping backbone cannot handle occlusion case, the results produced by our model might have artifacts. As shown in Fig. 8, when the target face is wearing glasses, the glasses can barely be recovered from the bad result generated by DFL [46]. This limitation may be alleviated by providing a parsing map to copy these areas from the target image.

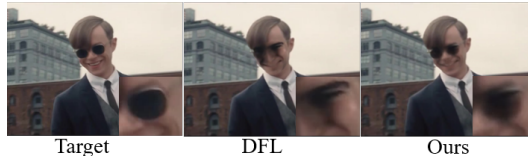

Figure 8: Failure Case. If pre-stage identity swapping backbone fails on the occlusion case, our model can hardly recover the lost details.

## 4.5 Forgery Detection

We Conduct binary detection on two video classification baselines: I3D [8] and TSN [63] on the hidden set provided by DPF-1.0 [25]. We trained the baselines on four manipulated datasets of FF++ [53] produced by DeepFakes [17], Face2Face [58], FaceSwap [18], and NeuralTextures [59]. Then, we add 100 manipulated videos produced by our method to the training set. All detection accuracies are improved with the addition of our data. Please refer to the supplementary material for details.

## 5 Conclusion

In this work, we provide a new identity swapping algorithm with large differences in appearance for face forgery detection. We proposed to formulate appearance mapping as an optimal transport problem and formulate it in both latent and pixel space. A relighting generator simulates the optimal transport plan estimation in the latent space. A segmentation game is further developed to refine the solution in pixel space. Extensive experiments demonstrate the significant superiority of our method over state-of-the-art methods.

# 6 Broader Impact

Deepfake refers to synthesized media in which a portrait of a person in real media is replaced by that of someone else. Deepfakes have been widely applied in the digital entertainment industry, but they also present potential threats to the public. Identity swapping is an approach to produce Deepfakes and is also the research direction of this paper. Given the sensitivity of Deepfakes and their potential negative impacts, we further discuss the potential threats and the corresponding mitigation solutions with respect to our work.

## 6.1 Potential Threats to the Public

Although the Deepfake was originally invented for social good, such as digital entertainment [69, 71, 62, 58, 44, 32], it may also be used for malicious proposes, thus exerting substantially negative impacts on individuals, institutions, and society.

**Political threats.** In the political arena, Deepfakes may be used to impact elections around the world or to foment political controversy. Malevolent Deepfakes may be part of fake news, which has the potential to undermine the political system, especially during an election year.

**Disinformation attacks.** The vivid Deepfake may be adopted as false evidence to mislead and incite the public. For example, social media postings with Deepfakes may maliciously create controversy over a sensitive topic.

**Identity theft.** Deepfakes facilitate the crime of obtaining the identity information of another person to conduct a transaction, such as financial fraud. For example, the Deepfake enables a criminal to pretend to be a manager and ask employees to send money. At the same time, powerful identity swapping applications can pose security problems when facial recognition or face forgery detection algorithms fail to detect manipulated faces.

**Celebrity pornography.** Making fake pornography has been a common threat, where faces of victims are swapped with those of porn stars. With today's technology, the identity swapping can be done after obtaining a set of social photos of the victims. In light of this, Deepfakes can lead to bullying or harassment, which may bring major psychological burdens and legal consequences to people.

Given these potential negative effects, we need a clear solution to limit the use of Deepfakes in order to reduce the threats to public safety and to protect people's rights.

## 6.2 Ways to Prevent the Harms and Our Contributions

There are several effective solutions to alleviate the aforementioned concerns.

**Keep up with the latest Deepfake technologies.** Understanding the characteristics of various Deepfake synthesis algorithms is crucial for the development of Deepfake detection algorithms. In order to facilitate tracking the latest technologies, common standards can be developed for summarizing and recording technology advances. Recent work adequately collates and summarizes latest Deepfakes developments. Such work will greatly help researchers to better understand current developments and develop new strategies to detect Deepfakes.

**Collect Deepfakes data.** Recent studies have shown that Deepfakes synthesized by different algorithms can be used as training data to improve the performance of detection algorithms. It is highly suggested that researchers actively expand the latest Deepfakes imagery into training data. These data need to be properly managed. For instance, non-profit organizations could be established to manage and maintain these data.

**Develop detection algorithms.** Using detection algorithms to automatically identify Deepfakes is a primary way to reduce their negative impact. Government agencies and related organizations should actively support the development of detection algorithms.

**Legislation.** Given the enormous potential harm of Deepfakes, there is an urgency to start the legislation process to regulate Deepfakes. The law should standardize the allowed usage of Deepfakes and the consequences of illicitly using Deepfakes to commit crimes.

**Our contributions to the community.** Apart from proposing an efficient algorithm to advance the development of generative models, we are also striving to mitigate the harm caused by Deepfakes: (1) We are building a new Deepfake dataset (synthesized by our algorithm) to advance the state of the art in Deepfake detection algorithms. (2) We are committed to support the development of Deepfake detection algorithms in any way, including but not limited to summarizing the latest Deepfake algorithms and developing novel detection algorithms. Please refer to our project page for the latest progress.

## Acknowledgment

This work is partially funded by Beijing Natural Science Foundation (Grant No. JQ18017) and Youth Innovation Promotion Association CAS (Grant No. Y201929).

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
