[Supplementary Material]

# ——SUPPLEMENTARY MATERIAL——
# AOT: Appearance Optimal Transport Based Identity Swapping for Forgery Detection

**Hao Zhu** [1,2,*] **Chaoyou Fu** [1,3,*]**, Qianyi Wu** [4]**, Wayne Wu** [4]**, Chen Qian** [4]**, Ran He** [1,3,†]
[1] NLPR & CEBSIT & CRIPAC, CASIA [2] Anhui University
[3] University of Chinese Academy of Sciences [4] SenseTime Research
haozhu96@gmail.com {chaoyou.fu,rhe}@nlpr.ia.ac.cn
{wuqianyi,wuwenyan,qianchen}@sensetime.com

## Appendix

## A    Implementation Details

### A.1    Data Preparation

The training data only consists of the target faces and the reenacted faces. The target faces are directly extracted from the original FF++ [9] (900 videos) and DPF-1.0 [4] (10 identities). Then, we leverage DFL [8] and FSGAN [6] to produce the reenact faces using identities that are not existed in the target faces. Our quantitative experiments are conducted on the remaining videos in FF++ and DPF-1.0.

Then, we first detect 106 facial landmarks of each video. Then, we crop the face area and resize them into 256*256 resolution. To obtain the PNCC and normals, we use 3DDFA [15] to estimate the 3D mesh of each face, and render the mesh and corresponding PNCC and normals codes to images via a neural renderer [5].

### A.2    Training Strategies

We use PyTorch [7] to implement our model. In the training phase, our model is trained with 200K iterations on two NVIDIA1080Ti GPUs, where the batch size = 16. We use Adam optimizer for relighting generator with $\beta1 = 0.5$, $\beta2 = 0.999$, weight decay = 0.0002, and RMSprop optimizer for Mix-and-Segment Discriminator (MSD), $\Omega$, $\Psi$ with beta = 0.9. The learning rates of both the Adam and the RMSprop optimizers are set to 0.0002. In $\mathcal{L}_{total}$, we set $\lambda_1$=120, $\lambda_2$=1, $\lambda_3$=90, and $\lambda_4$=1. The full training algorithm is summarized here 1.

### A.3    Network Architectures

The full architecture as shown in Fig. S1.

---

**Algorithm 1** Training algorithm.

---

**Require:** $\{X_r\}^N, \{X_t\}^N$;
**Require:** Initialize $\Omega_i, \Psi_i, PerceptualEncoder, Decoder$, and $MSD$ with $\theta_i, \omega_i, \alpha, \beta, \gamma$ respectively.

1: **while** not converged **do**
2:     Sample mini-batch $\{x_r\}$
3:     Sample mini-batch $\{x_t\}$
4:
5:     // Forward: Encoder
6:     $F_{X_r}^1, F_{X_r}^2, F_{X_r}^3, F_{X_r}^4 \leftarrow \text{PERCEPTUALENCODER}(\{x_r\})$
7:     $F_{X_t}^1, F_{X_t}^2, F_{X_t}^3, F_{X_t}^4 \leftarrow \text{PERCEPTUALENCODER}(\{x_t\})$
8:
9:     // Update: NOTPE
10:     **for** $i = 1, 2, 3$ **do**
11:         **for** $j = 1, ..., n_i$ **do**
12:             Sample $v_r^j \leftarrow F_{X_r}^i$
13:             Sample $v_t^j \leftarrow F_{X_t}^i$
14:             $g_{\omega_i} \leftarrow \nabla_{\omega_i}[\frac{1}{m}\Psi_i(\Omega_i(v_r^j|X_r, X_t)) - \frac{1}{m}\Psi_i(v_t^j)]$
15:             $\omega_i \leftarrow \omega_i + \alpha \cdot \text{RMSProp}(\omega_i, \text{x})$
16:             $\omega_i \leftarrow \text{CLIP}(\omega_i, -c, c)$
17:         **end for**
18:     **end for**
19:     **for** $i = 1, 2, 3$ **do**
20:         **for** $j = 1, ..., n_i$ **do**
21:             Sample $v_r^j \leftarrow F_{X_r}^s$
22:             $g_{\theta_i} \leftarrow -\nabla_{\theta_i}\frac{1}{m}\Psi_i(\Omega_i(v_r^j|X_r, X_t))$
23:             $\theta_i \leftarrow \theta_i + \alpha \cdot \text{RMSProp}(\theta_i, \text{x})$
24:         **end for**
25:     **end for**
26:
27:     // Forward: NOTPE
28:     **for** $i = 1, 2, 3$ **do**
29:         $F_Y^i \leftarrow \Omega_i(F_{X_r}^i, X_r, X_t)$
30:     **end for**
31:
32:     // Forward: Decoder
33:     $Y_{t,t} \leftarrow \text{DECODER}(F_{X_r}^1, F_{X_r}^2, F_{X_r}^3, F_{X_r}^4)$
34:     $Y_{r,t} \leftarrow \text{DECODER}(F_Y^1, F_Y^2, F_Y^3, F_{X_r}^4)$
35:
36:     // Update: MSD
37:     $M_r \leftarrow \text{Random Mask Generator}()$.
38:     $Y_{mix} \leftarrow \text{MIX}(Y_{t,t}, Y_{r,t}, M_r)$
39:     $g_\gamma \leftarrow \nabla_\gamma[\frac{1}{m}][M_r * MSD(Y_{mix})] - \frac{1}{m}[(1 - M_r) * MSD(Y_{mix})]$
40:     $\gamma \leftarrow \gamma + \alpha \cdot ADAM(\gamma)$
41:
42:     // Update: Perceptual Encoder, Decoder
43:     $g_\beta \leftarrow \nabla_\beta[\frac{1}{m}]Loss(Y_{t,t}, Y_{r,t}, X_t)$         ▷ Total Loss
44:     $\beta \leftarrow \beta + \alpha \cdot ADAM(\beta)$
45: **end while**

Figure S1: The detailed pipeline of our proposed model.

## B  Compared Baseline

### B.1  Face Swapping Methods

**DeepfaceLab.**  DeepfaceLab (DFL) [8] requires to retrain the model for different source identities. It means we need to train the DFL model different videos respectively. It should be clear that, DFL provides lots of options to tune the results. In practice, we use the options reported in Table S1.

**FSGAN.**  FSGAN [6] is a landmark-guided subject agnostic method. We leverage the latest models provided by authors.

### B.2  Appearance Transfer Methods

**Poisson Blending.**  Poisson Blending is a classical image harmonization method.  We use the OpenCV implemented version, and set the flag=cv2.NORMAL_CLONE.

**Deep Image Harmonization (DIH) [13].** [3]   DIH is a deep learning based image harmonization method and it can capture both the context and semantic patterns of the images rather than hand-craft features.

Table S1: Options of DeepFaceLab.

| Training Options | | | | Merging Options | |
|---|---|---|---|---|---|
| name | choice | name | choice | name | choice |
| resolution | 224 | gan_power | 0.0 | mask_mode | learned |
| face_type | f | true_face_power | 0.0 | erode_mask_modifier | 5 |
| models_opt_on_gpu | True | face_style_power | 0.0 | blur_mask_modifier | 5 |
| archi | dfhd | bg_style_power | 0.0 | motion_blur_power | 0 |
| ae_dims | 256 | ct_mode | None | output_face_scale | 1 |
| e_dims | 64 | clipgrad | False | color_transfer_mode | rct |
| d_dims | 64 | pretrain | False | sharpen_mode | none |
| d_mask_dims | 22 | autobackup_hour | 0 | blursharpen_amount | 0 |
| masked_training | True | write_preview_history | True | super_resolution_power | 1 |
| eyes_prio | False | target_iter | 0 | image_denoise_power | 0 |
| lr_dropout | False | random_flip | True | bicubic_degrade_power | 0 |
| random_warp | True | batch_size | 4 | color_degrade_power | 0 |

**Style Transfer for Headshot Portraits (STHP) [10].** [4]    STHP allows users to easily produce style transferred results. It transfers multi-scale local statistics of an reference portrait into another.

**WCT$^2$.** [5]    WCT$^2$ is a state-one-the-art photorealistic style transfer method. We use the option unpool = 'cat5' version, and the pretrained models.

# C    Additional Experiments

## C.1    Noise Analysis

Furthermore, we verified our results with photo forgery methods: noise analysis, error level analysis, level sweep, luminance gradient [6]. As shown in Fig. S2, ours framework reduces the noises (Fig. S2 (a, b)) and preserves the appearance with target images (Fig. S2 (c, d)).

Figure S2: Noise analysis with photo forensics algorithms. Our method can not only reduce the noises (a,b), but also better preserve appearances. (c,d).

Figure S3: The mixed results.

## C.2 Results of Mix-and-Segment Discriminator

We provide more results of the mixed results. As shown in Fig. S3, we mix the target faces and the swapped faces using the mix mask. It is difficult to find the real patch and the fake patch.

## C.3 Feature Visualization

To give intuitive results, we visualize the features at different scales by using PCA to reduce the dimensions of them to 3-dimensional vectors.

In the latent space the pixel distributions are more balance under different lighting conditions, as shown in Fig. S4.

Figure S4: Visualization of the features at different scales.

Table S2: Inference speed comparison

| Methods | FPS |
|---------|--------|
| Poisson | 3.891 |
| DIH | 1.247 |
| STHP | 1.686 |
| WCT | 2.817 |
| AOT (ours) | 12.821 |

## C.4 Speed Comparison

Furthermore, as reported in Table S2, our framework achieves the highest FPS compared with other related methods, which means our method introduces the minimum computational burdens. All experiments conducted on Ubuntu16.04 with an Intel i7-7700K CPU and a Nvidia 1060 GPU.

## C.5 Forgery Detection

Binary detection accuracy of two video classification baselines: I3D [1] and TSN [14] on the hidden set provided by DeeperForensics-1.0 [4].

We trained the baselines on four manipulated datasets of FF++ [9] [9] produced by DeepFakes [2], Face2Face [11], FaceSwap [3], and NeuralTextures [12]. (Green bars). Then, we add 100 manipulated videos produced by our method to the training set. All detection accuracies are improved with the addition of our data. (Blue bars).

Figure S5: Forgery Detection Results.

| Target | DFL | +Poisson Blending | +DIH | +STHP | +WCT$^2$ | Ours |
|--------|-----|-------------------|------|-------|----------|------|

Figure S6: Comparison results with DFL.

| Target | FSGAN | +Poisson Blending | +DIH | +STHP | +WCT$^2$ | Ours |
|--------|-------|-------------------|------|-------|----------|------|

Figure S7: Comparison results with FSGAN.

## Footnotes

[3]DIH: https://github.com/wasidennis/DeepHarmonization

[4]STHP: https://people.csail.mit.edu/yichangshih/portrait_web/

[5]WCT$^2$: https://github.com/clovaai/WCT2

[6]https://29a.ch/photo-forensics/