[Reviews · NeurIPS 2020]

Review 1

Summary and Contributions: The paper present a post-processing method for "Face Swapping" algorithms that reduces the discrepancies in lighting and color distribution that many algorithms usually create between their output and the target image. The major contributions are: - Formulation of the post-processing step as an Optimal Transport problem based on the distribution of features in image space. The Wasserstein algorithm is used in an unusual (=novel) way to solve the problem. - A novel conditional GAN discriminator that learns to detect blending artifacts. - The output of the method seems to reliably produce plausible results, in contrast to the face-swapping techniques it is supposed to serve as a post-process for.

Strengths: - In principle this method should be applicable to any face-swapping - In contrast to the results from the original face-swapping techniques, the results from this method always look flawless to the human - The way that the OT/Wasserstein method is used to perform image-to-image translation seems very clever and novel to me, especially since it incorporates not only pixel color, but also more abstract image - The evaluation is extensive and contains quantitative (metrics) and qualitative (user study) - The "mix-and-segment" discriminator seems to be novel as well, at least for the application of face-swapping

Weaknesses: - The authors never clearly state (neither in the manuscript, nor in the supplemental material) which information is available to the network at *test* time. I do not understand if their system is trained once on a large corpus of face swaps and can then be applied to arbitrary reenactment results or if it is in some way specific to a certain subject or setting. Also I do not understand whether the target image of a swap also needs to be supplied at test time. It is not clear if the NOTPE network is retrained at test time. This lack of clarity is the biggest complaint I have. - There is no discussion of limitations and no failure cases are shown. - In terms of exposition, this submission does not clear the bar of what I would expect from a NeurIPS publication (see below).

Correctness: Overall the claims made in the paper appear to be substantiated. Minor flaws: - Line 182 claims that Wasserstein GAN "minimizes the Wasserstein distance between two images". This statement is wrong: Wasserstein GAN minimizes the Earth-Mover distance between two probability distributions. - In part (b) of Table 1 it is not clear what the "scores" mean and section 4.2.2 does not clearly say it either. I was able to infer that probably the scores represent the rate at which users picked the result of the presented method as the "better" one, but I think the text should explicitly say that.

Clarity: The overall structure of the paper is good. Linguistically however, the paper can in my opinion not stay the way it is: - Many sentences are not grammatical, which makes them hard to read. I only list a small subset of examples: Line 5: "appearance gaps mainly come from the large discrepancies [...], widely exist*ed* in real-world scenarios" Lines 34 - 36: This sentence lacks structure. Also in the following sentence it seems as if "this goal" refers to having "obvious artifacts and inaccurate skin colors or illuminations". Line 46: "which causes the inability to fine-grained appearance translation" Line 61: The sentence starting here does not have a predicate. Line 77: The sentence starting at the beginning of this line does not have a predicate. Line 91: "the realistic of the output image has been neglected." Lines 134 - 136: This sentence would be easier to understand if conjunctions would be used in-between its parts. The same applies to Line 228. Line 156: Instead of "Specificity" (noun) it should say "Specifically" (adverb) Line 262: Sentence not grammatical. Line 265: "under face extreme conditions". Incorrect order of words. Lines 267 - 268: Sentence not grammatical. - Some vocabulary is used in misleading ways that make sentences confusing. Select examples: * Lines 32 & 33: An example presented in Figure 1 is called an "intractable case". At the same time the authors claim that their method handles this case well, which contradicts the meaning of the word "intractable". * Line 49: Instead of "we expose" it should say "we propose". * Line 37: Here and in other places the authors write of "solving a mapping". I think they mean "finding a mapping". "Solving a mapping" may be informal, colloquial style that researchers use in spoken conversation, but it is inappropriate in a written publication. * Line 50: Similarly, the world "settle" is used in conjunction with "problem" several times. I am unable to figure out whether the authors mean "solve" or "formulate" here. * Line 53 & 143: Instead of "oriental information" (= "information from the Middle East") it should say "orientational information". * Line 105: "OT solves". No. OT is defined to *be* a problem (see sentence before). It does not "solve" a problem. * Line 113: "Inspired by the wide range of applications of OT, however, the OT map has historically been difficult to compute". This suggests that after/because OT had become wide-spread, its difficulty increased. This is clearly nonsensical. * Line 158: The word "identical" confused me quite a lot (because it sounded like there was a redundancy in the feature space), until I figured out that it should really say "identity" here. - Sometimes adjectives/adverbs with very subjective meaning are used, which is a no-go in a scientific publication. Some examples: Line 10: "a relighting generator is elaborately designed". Scratch "elaborately" Line 27: "unprecedented success of deep learning" (scratch "unprecedented") & "face swapping further makes gratifying progress" (scratch "gratifying") The given examples are by no means exhaustive; almost every sentence in the paper contains at least one flaw. In many cases these mistakes can be worked around by the reader, but in some cases they make it very hard to understand what the authors want to say. Minor problems: - In my opinion, the words "source" and "target" in the sentence in line 20 should be swapped. - The term "segmentation game" is used as if it was a known term from the literature, when in fact the authors have created it themselves. At least they do not provide a citation for it... - Caption of Figure 1: Instead of "Comparisons results" write either "comparison results" or just "Comparisons" - Line 204: There are three asterisks ("***") that do not seem to mean anything. - There is a Greek letter in the bottom right portion of Figure 2 that I think should be Psi. If it's not Psi, it was never defined anywhere in the paper.

Relation to Prior Work: Yes, section 2 points out the differences between the submission and the previous work.

Reproducibility: Yes

Additional Feedback: Yes. There is pseudocode for the training loop and a detailed sketch of the networks involved in the supplemental material. -- The rebuttal addresses some of my points. I hope the authors can come up with a more polished paper in terms of language. I would still like to accept the paper.


Review 2

Summary and Contributions: The authors propose to solve the problem of inconsistent face swapping results when there are large appearance gaps between the source and the target images, including illuminations and skin colors. The goal is to model the complex appearance mapping so as to transfer fine-grained appearances adaptively with identical traits preservation. Promising results are also achieved compared to DeepfaceLab and FSGAN.

Strengths: The idea to achieve appearance transfer on face swapping by first transferring the latent features and then use a face decoder to generate the final result is somewhat novel.

Weaknesses: 1. Eq. (3) and Eq. (4) seem to be different. In Eq. (3), the authors aim to solve an optimal transport plan between the feature histograms of $$F_{X_r}^i$$ and $$F_{X_t}^i$$. In optimal transport, such an optimal transport plan not only transports the source distribution to the target distribution, but also minimizes the transportation cost. In Eq. (4), the Wasserstein distance between the mapped distribution and the target distribution is minimized. In this way, the mapping function $$\omega$$ maps the source distribution to the target distribution, but the transportation cost is not minimized. 2. Sec. 3.2 is not easy to follow. What is the relation between the proposed Mix-and-Segment Discriminator and the `mixup' in Sec. 3.7 of [1]. Besides, the empirical comparison against WGAN should be added. 3. Metrics including `verification', `pose', `landmarks' should be added as they are all used in the two baseline face swapping methods. 4. Time complexity should also be compared with the competitors. [1] Zhang, H., Cisse, M., Dauphin, Y. N., & Lopez-Paz, D. (2018, February). mixup: Beyond Empirical Risk Minimization. In International Conference on Learning Representations.

Correctness: Yes

Clarity: Yes

Relation to Prior Work: Yes

Reproducibility: Yes

Additional Feedback:


Review 3

Summary and Contributions: The authors propose AOT, a method to improve face swapping results by addressing the changes in lighting and skin tones. The authors extend the idea of Optimal Transport Model by introducing appearance related features and apply it in both latent and pixel space. The authors introduce NOTPE to adapt OT idea into neural network training. The method produces plausible results qualitatively and is quantitatively better than other methods compared.

Strengths: 1. Formulation of the problem. The authors address the latent feature transformation problem using a NOTPE. They address the similarity term in the image space using adversarial training. The overall network design is solid and well established. 2. Plausible results. The swapping results shown in fig1 are clearly better than other methods, and artifacts caused by changes of lighting and skin colors are barely visible.

Weaknesses: 1. Related work. This work replies on the previous face swapping methods [31, 29], and mainly takes their results as input and corrects the artifacts. It is worth comparing to other state-of-arts fully automatic or end-to-end methods, and justify the reason why a two-stage computation is preferred here. 2. Loss of details. The network relies on an encoder-decoder which is prone to loss of high-frequency details of the images. As shown in the supplemental video from 4:41, the results look blurry compared to the input. The authors would need to justify their work does not worsen the results from previous methods.

Correctness: The claims and methods are correct in this paper.

Clarity: Yes. The reviewer found the following points which might get the authors' attention: 1. line 204: extra *** 2. in supp video: 'Neutral network' -> 'Nueral network'

Relation to Prior Work: Yes.

Reproducibility: Yes

Additional Feedback:


Review 4

Summary and Contributions: This paper proposed an enhanced face-swapping framework via optimal transport. After getting the initial swapped face using any previous swapping method, the system will refine its color and illumination using a Relighting Generator. This generator has a modified UNet structure, with Neural Optimal Transport Plan Estimation (NOTPE), to transport features in skip connections. It also employs a Mix-and-segment Discriminator to enforce realisticity. The proposed method outperforms the state-of-the-art ones on both qualitative, metric, and user evaluation.

Strengths: * AOT applies optimal transport (OT), an emerging technique, in improving face-swapping results. However, it recognized the problem of conventional OT when applying to this problem. Hence, it proposed a modified version, which is included inside the deep generator. AOT shows much more realistic swapped images compared with the previous methods as well as the conventional OT. * AOT exploits 3D face estimation to improve further generated results. * Mix-and-segment Discriminator is an interesting and effective component. * AOT outperforms the state-of-the-art ones on both qualitative, metric, and user evaluation.

Weaknesses: * Some results look less like the source face. The authors should add analyses on identity preservation, which is important in face-swapping. * In equation (7), the second term is not for style loss. The authors should find another name for it. * Minor issues: - Section 4.1: redundant "***". Also, Poisson Blending does not show "the ghosting" effect in Fig. 4, as mentioned in L206. - Typos: "assess" (L231), "lose" (L237). - More challenging examples, such as cross-gender swapping should be added.

Correctness: Yes.

Clarity: Yes

Relation to Prior Work: Yes

Reproducibility: No

Additional Feedback: The rebuttal addressed my comments. I believe this paper is good for publication. While there are some ethical concerns, I still lean toward accepting this paper. To improve deepfake detection, we need to understand the ability of deepfake methods, and deepfake generation studies are essential. This work will raise awareness of this problem. and increase awareness on this problem,

[Author Response · NeurIPS 2020]

**Response to reviewers (*Submission ID #256*)**

We thank all reviewers for their efforts and constructive comments. Below we give our detailed responses.

**To Reviewer #1: A1. The "test time" of our model.** Thanks for your comments. ***Training settings:*** Our model is

trained on a large corpus and it can be applied to the identities that are not presented in the training set. Principally, the

larger training set, the better generalization. ***Whether the target image is required:*** Yes, the target image is leveraged

as a condition to realize the face swapping. ***Retrain or not:*** We do not retrain any modules in the testing phase. These

details will be updated in the revised version.

**A2. Limitations and failure cases.** If a face swapping backbone cannot handle

occlusion cases, the results produced by our model might have artifacts (see Fig. 1).

These will be added in the revised version.

**A3. Minor flaws.** Thanks for your careful review. We have revised all the typos and

we will invite a native speaker to proofread our manuscript. The related references

Target    DFL    Ours
Figure 1: Failure cases.

have been added in the revised version. The scores in Table 2 (b) represent "the rate at which users picked the result of

the presented method" as you inferred. We will make it clear in the revised version.

**To Reviewer #2: A1. The differences between Eq. (3) and Eq. (4).** Eq.(4) is derived from $W =$

$\sup_{\Psi \in \mathcal{F}^1}[\mathbb{E}_{v_r \sim \mathbb{V}_r}(\Psi(v_r)) - \mathbb{E}_{v_t \sim \mathbb{V}_t}(\Psi(v_t))]$, which is the dual form of Eq. (3). Hence, optimizing Eq. (4) will

minimize the transportation cost implicitly. The detailed proof of the equality of Eq. (3) and Eq. (4) will be provided in

the revised version. Additionally, $\Omega$ serves as the generator in WGAN. In WGAN's setting, both [1] and [2] agree that

the generator aims to solve the optimal transportation. Finally, we also visualize the transportation as shown in Fig. 6.

**A2. Relation between the "mixup" and the MSD.** The "mixup" is a part of MSD. It is denoted as "MixLayer" in

Fig. 2. As a data augmentation technique, mixup is originally proposed for empirical risk minimization. Differently, we

use the "mixup" operation to mix two images, forcing the generator to produce more realistic results (see R2A3).

**A3. Experiments on 'verification', 'pose', 'landmarks'.** We have conducted the relative experiments, as reported in

Table 1. Our method achieves comparable performance compared with pre-stage face swapping methods. It verifies

that, as a post-processing step for face swapping, our method does not degrade the performance of previous methods.

**A4. Empirical comparison against WGAN.** MSD is leveraged to distinguish which part of an image is "fake",

providing fine-grained information to guide the generator. The generator can produce smooth and well-blended results

under this supervision. However, WGAN takes the whole image as a condition, thus the detailed information may be

neglected. Experimental results in Table 2 verify the superiority of our method over WGAN.

**A5. Time complexity.** Please refer to supplementary material (C.4).

**To Reviewer #3: A1. Compare to other fully automatic or end-to-**

**end methods.** Firstly, FSGAN is a fully automatic method, and we

have already compared with it (Fig. 3, Fig. 4, and Table 1). In addition,

we cannot provide the comparison with the other SOTA end-to-end

method (FaceShifter [3]), since the official code is unavailable.

Table 1: Results of verification, pose, and landmark on both FF++ and DPF-1.0.

|  | verification↓ | | pose↓ | | landmark↓ | |
|---|---|---|---|---|---|---|
|  | FF++ | DPF | FF++ | DPF | FF++ | DPF |
| DFL | 0.231 | 0.243 | 3.161 | 3.940 | 3.073 | 3.289 |
| +Ours | 0.237 | 0.249 | 3.159 | 3.953 | 3.070 | 3.316 |
| FSGAN | 0.314 | 0.392 | 2.631 | 2.767 | 2.823 | 2.412 |
| +Ours | 0.317 | 0.389 | 2.638 | 2.762 | 2.831 | 2.409 |

**A2. Why a two-stage computation is preferred.** Due to the difficul-

ties of modeling the complex appearance mapping directly, one-stage

methods, which are required to process expressions, poses, and appear-

ances simultaneously, tend to fail under complex lighting conditions. Therefore, a refinement is necessary. Furthermore,

as emphasized by R1, as a post-processing model, our model can be easily concatenated with other reenactment models

to form a fully automatic face swapping pipeline.

**A3. Justify our model does not worsen the results from previous methods.** The proposed perceptual encoder

leverages 3D information to encode the semantic and geometric information, which is important to the generation. Also,

a UNet-like architecture can preserve the earlier information with the skip connections. Furthermore, the proposed

MSD can also improve synthesis performance. Although our model produces the blurry cases under some specific

conditions (Please refer to R1A2), extensive quantitative and qualitative results (Fig. 3, Fig. 4, and Table 1) on a

large-scale dataset verify the improvements over previous methods.

**To Reviewer #4: A1. Analyses on identity preservation.** As a post-processing

stage for face swapping, our method mainly focuses on refining the reenactment

results while not changing the generated identities of previous methods. Hence,

identity preservation mainly relies on previous reenactment methods. Experimental

Table 2: The evaluation on FF++.

|  | SSIM-e | SSIM-w | gram loss |
|---|---|---|---|
| WGAN | 0.8089 | 0.7315 | 0.013667 |
| Ours | **0.8301** | **0.7810** | **0.003578** |

results of "verification" on Table 1 verify that our method does not degrade the performance of the backbone methods.

**A2. Name of style loss.** We will use "appearance loss" in the revised version.

**A3. More challenging examples (such as cross-gender).** Please refer to the supplementary material Fig. S5 (bottom

row) and Fig. S6 (top row) for cross gender results. These figures also contain some results with large poses.

**[1]** Gulrajani et al. Improved Training of Wasserstein GANs, NeurIPS, 2017

**[2]** Lei et al. A Geometric View of Optimal Transportation and Generative Model, CAGD, 2019

**[3]** Li et al. FaceShifter: Towards High Fidelity and Occlusion Aware Face Swapping, CVPR, 2020


[Meta-Review · NeurIPS 2020]

This paper received overall positive reviews from four reviewers. The reviewers like the quality of the visual results, the extensiveness of the evaluation, and the novel idea of appearance transfer. Some concerns include clarity of presentation, no discussion of limitations, etc. From the perspective of technique merit, the consensus is that this paper shall be accepted. However, since the topic of this paper is regarding new approaches of generating synthetic faces, there are potential ethical concerns as indicated by some reviewers. As a result, we further send this paper to two NeurIPS ethical reviewers for additional comments. The ethical reviewer #1 stated that “the broader impact statement needs to be much stronger.”, for a number of reasons: “1) The paper positions itself as theoretical and therefore not posing a practical risk, but at the same time the paper does proposed a concrete algorithm and displays examples of significantly improved deepfakes. This is inconsistent. The authors should clarify. I agree no tool (or code for reproducibility is offered - which is a different kind of problem), but the algorithm seems clear enough someone could reproduce it with enough work. 2) The paper glosses over the very serious risks posed by deepfake techniques. The paper mentions security and privacy, but it doesn't mention manipulated media, misinformation, hoaxes and false news, fraud, defamation, etc.” The ethical reviewer #2 stated that “the broad impact statement is brief, and limited in scope. It claims that a potential positive impact is bringing deceased actors back to life by swapping their faces onto substitutes. However, this is a controversial idea as it could raise a legal action under the actor's right to publicity. But it is the discussion of ethical considerations that is a more serious shortcoming: it offers only a passing mention of security concerns and privacy harms. There is no mention of the widespread research and public discussion of the risk of these tools used in harassment, impersonating public figures, and revenge porn (see the scholarship of Danielle Citron among others). These are direct harms that far exceed privacy, and should be raised under the categories of potential harms in Section 4A. 4): "could it be used to impersonate intimate relations for the purpose of theft or fraud? Could it be used to impersonate public figures to influence political processes?" These are not theoretical concerns, but harms that already occur with these kinds of systems in the world today.” In light of the ethical concerns, the AC recommends a conditional acceptance of this paper. That is, this paper can be accepted after the authors make substantial improvement on the broad impact statement, especially in thoroughly addressing the concerns raised by the two ethical reviewers. For example, the ethical reviewer #1 mentioned that “the potential mitigations include the creation of methods for detecting deepfakes and of datasets to help researchers train deepfake detectors.” The authors should address the issue that “The paper mentions the method proposed can be used to generate more challenging deepfake datasets, but the authors don't share any data as far as I can tell.” The ethical reviewer #2 mentioned “This is an emerging area of research, and as the authors acknowledge, there is potential in this work to strengthen forgery detection algorithms. The paper should emphasize the serious risk of harm, and how this tool can be used to address those harms first and foremost. If it cannot propose any way to prevent these harms, but only to "improve" the ability for face swapping using the Appearance Optimal Transport model, then this paper brings undue risk of harm.” Furthermore, the ethical reviewer #2 suggested that “given the many news articles and research discussions about the harms of face swapping and deep fakes over the last two years, it would strengthen this paper to engage with those risks in detail. At present, it really only assesses this space as an optimal transport problem as a way to improve performance, rather than seriously contending with the serious harms of these tools being widely available (and considering who is most at risk).” ******************************* Note from Program Chairs: The camera-ready version of this paper has been reviewed with regard to the conditions listed above, and this paper is now fully accepted for publication.